# Evaluation of Amyloid β_42_ Aggregation Inhibitory Activity of Commercial Dressings by A Microliter-Scale High-Throughput Screening System Using Quantum-Dot Nanoprobes

**DOI:** 10.3390/foods9060825

**Published:** 2020-06-24

**Authors:** Masahiro Kuragano, Wataru Yoshinari, Xuguang Lin, Keiya Shimamori, Koji Uwai, Kiyotaka Tokuraku

**Affiliations:** Graduate School of Engineering, Muroran Institute of Technology, Muroran 050-8585, Japan; gano@mmm.muroran-it.ac.jp (M.K.); ysnrwtr0629@icloud.com (W.Y.); linxuguang4000@163.com (X.L.); 19041040@mmm.muroran-it.ac.jp (K.S.); uwai@mmm.muroran-it.ac.jp (K.U.)

**Keywords:** Alzheimer’s disease, Amyloid β, amyloid β aggregation inhibitor, quantum dot, soy sauce

## Abstract

The aggregation and accumulation of amyloid β (Aβ) in the brain is a trigger of pathogenesis for Alzheimer’s disease. Previously, we developed a microliter-scale high-throughput screening (MSHTS) system for Aβ_42_ aggregation inhibitors using quantum-dot nanoprobes. The MSHTS system is seldom influenced by contaminants in samples and is able to directly evaluate Aβ_42_ aggregation inhibitory activity of samples containing various compounds. In this study, to elucidate whether the MSHTS system could be applied to the evaluation of processed foods, we examined Aβ_42_ aggregation inhibitory activity of salad dressings, including soy sauces. We estimated the 50% effective concentration (EC_50_) from serial diluted dressings. Interestingly, all 19 commercial dressings tested showed Aβ_42_ aggregation inhibitory activity. It was suggested that EC_50_ differed by as much as 100 times between the dressings with the most (0.065 ± 0.020 *v*/*v*%) and least (6.737 ± 5.054 *v*/*v*%) inhibitory activity. The highest activity sample is traditional Japanese dressing, soy sauce. It is known that soy sauce is roughly classified into a heat-treated variety and a non-heat-treated variety. We demonstrated that non-heat-treated raw soy sauce exhibited higher Aβ_42_ aggregation inhibitory activity than heat-treated soy sauce. Herein, we propose that MSHTS system can be applied to processed foods.

## 1. Introduction

One of the problems facing an aging society is the increase of patients with dementia. While various diseases are known to cause dementia, Alzheimer’s disease (AD) in particular, accounts for the majority of cases [1,2,3]. Four AD drugs approved in Japan, donepezil, galantamine, rivastigmine, and memantine, only function by delaying the progression of pathological conditions by temporarily enhancing neurotransmission, and are not fundamental therapeutic agents [4]. The amyloid cascade hypothesis notes that AD is caused by the aggregation and accumulation of 38 to 43 residues of the amyloid β (Aβ) peptide excised from amyloid precursor protein in the brain [5,6,7,8]. Recently, Biogen and Eisai reported that a patient’s cognitive decline had been blunted in clinical trials using antibodies, aducanumab, that bind specifically to Aβ aggregates [9]. However, in March of 2019, a phase III clinical trial of aducanumab was halted because of insufficient evidence to support its effect in AD [10]. In October of 2019, both companies announced that they would apply for a new drug application of aducanumab to the U.S. Food and Drug Administration in 2020, as the effect was confirmed in some patients who received the drug at a high dose. However, these events remind us of the difficulties in developing AD therapeutics. Therefore, attention is now focused on AD prevention and treatment schemes that target the aggregation and accumulation of Aβ. There is currently a global search for candidate substances that can inhibit Aβ aggregation. Since the aggregation and accumulation of Aβ begins several decades before the expression of AD [11], long-term prevention with functional foods may be more effective than treatment with therapeutic medicine.

Rosmarinic acid (RA) is a polyphenol found in abundance in plants of the *Lamiaceae* such as rosemary, perilla, and lemon balm. RA is a known inhibitor of Aβ aggregation [12,13]. Its Aβ aggregation inhibitory activity was examined using AD model mice and its safety was confirmed in human studies using lemon balm extract [14,15]. Among many other polyphenols, curcumin, which is found in turmeric, is also a famous Aβ aggregation inhibitor [12]. Further, it was reported that importance of functional foods on AD. The extract obtained from miso, a traditional fermented dressing in Japan, suppresses Aβ-induced neuronal damage [16]. Hsu et al., reported that nattokinase degraded amyloid fibrils [17]. Thus, the use of functional foods has attracted attention as a possible AD countermeasure. However, it is technically very difficult to evaluate plant extracts and processed foods as these include various impurities. In general, the Thioflavin T (ThT) method has been used to evaluate Aβ aggregation inhibitory activity of various substances [18]. ThT emits fluorescence when bound to amyloid fibrils. In this method, the level of Aβ aggregation is measured from the fluorescence intensity of ThT. However, the excitation and emission wavelengths of ThT are 455 and 490 nm, respectively, so they compete with the absorption wavelengths of many natural substances. Therefore, the ThT method is unsuitable to evaluate food samples that contain various contaminants. A method of directly observing Aβ aggregates with a transmission electron microscope (TEM) is widely used. Because it is necessary to dry the Aβ aggregates sample when preparing, the observation under physiological conditions is difficult. Further, the amount of aggregates is biased depending on the field of view even in the same sample, suggesting that there is a problem in quantitative. In addition, the ThT and TEM method generally require several steps for sample preparation and observation, and it is difficult to analyze a large amount of the sample at one time. In other words, previous conventional method could not perform accurate and quick high throughput quantitative analysis.

Previously, we succeeded in real-time imaging of the Aβ_42_ aggregation process with a fluorescence microscope using a quantum dot (QD) nanoprobe and developed a microliter-scale high-throughput screening (MSHTS) system for Aβ_42_ aggregation inhibitors by applying this imaging method [19,20]. The MSHTS system has some advantages: (1) only a small sample volume of 5 μL is required, (2) high-throughput analysis uses a 1536-well plate, and (3) filter effects due to contaminants in the sample are avoided because the amount of Aβ_42_ aggregates is quantified from standard deviation (SD) value estimated from the variation in fluorescence intensity of each pixel of obtained images and the emission wavelength of QD605 does not overlap with the absorption of almost natural products [20,21]. Thus, the MSHTS system can evaluate the magnitude of inhibitory activity for Aβ_42_ aggregation as EC_50_ values. Before, we evaluated the Aβ_42_ aggregation inhibitory activity of 52 spices using this method and demonstrated that the herb-based spices of the *Lamiaceae* family exhibited high Aβ_42_ aggregation inhibitory activity [20]. Then, we found that the activity of boiling water extracts of 11 seaweeds was higher than that of ethanolic extracts and revealed that Aβ_42_ aggregates morphology was affected with seaweed-derived polysaccharide including in boiling water extracts [22]. Further, we recently developed an automated MSHTS system to evaluate larger numbers of samples at once [21]. Screening 504 plant extracts collected in Hokkaido, Japan, we found that Geraniales and Myrtales within Rosids showed high Aβ_42_ aggregation inhibitory activity. Thus, MSHTS system is useful for quantitative evaluation of Aβ_42_ aggregation inhibition ability of various natural products. However, it is unclear whether MSHTS system can evaluate Aβ_42_ aggregation inhibitory activity in foods including various natural substances with many impurities. In this study, to elucidate whether the MSHTS system is applied to processed foods such as salad dressings, including soy sauces, we evaluated Aβ_42_ aggregation inhibitory activity of dressings using the MSHTS system. We found that all tested commercial dressings showed Aβ_42_ aggregation inhibitory activity despite there were differences in their activities. Especially raw soy sauce showed the highest inhibitory activity among the tested samples. These results suggest that the MSHTS system is a powerful and useful tool that is expected to be applied and developed in various processed foods.

## 2. Materials and Methods

### 2.1. Materials

Human Aβ_42_ (4349-v, Peptide Institute Inc., Osaka, Japan) and Cys-conjugated Aβ_40_ (23519, Anaspec Inc., Fremont, CA, USA) kits were purchased commercially. Twenty different commercially available salad dressings and soy source brands were purchased from Japanese companies (Kewpie, Sameura Foods, Sanyo Coffee Foods, Shiranukacho Shinko Kosha, Shinshu Shizen Okoku, Seijo Ishii, Taiyo Sangyo, Tsukiboshi Foods, Nihon Syoyu Kogyo, Big Chef, Pure Foods Toya, Yamada Bee Farm, Riken Vitamin, H+B Life Science) using catalog shopping in June 2016.

### 2.2. Preparation of QDAβ Nanoprobe

The QDAβ nanoprobe was prepared using QD-PEG-NH_2_ (Qdot^TM^ 605 ITK^TM^ Amino (PEG) Quantum dot; Q21501MP, Thermo Fisher Scientific, Waltham, MA USA) according to our previous reports [19,20,21,22]. The QDAβ nanoprobe was prepared by first reacting 10 µM QD-PEG-NH_2_ with 1 mM sulfo-EMCS (22307, Thermo Fisher Scientific, Waltham, MA, USA) in PBS (phosphate-buffered saline) for 1 h at room temperature. QDAβ concentration was determined by comparing absorbance at 350 nm to that of unlabeled QD-PEG-NH_2_.

### 2.3. Estimation of EC_50_ by the MSHTS System

The EC_50_ values of various dressings were determined by a modified MSHTS system, as was described in our previous reports [20,21,22]. More specifically, various concentrations of each dressing, 30 nM QDAβ, and 30 μM Aβ_42_ in PBS containing 5% EtOH and 3% DMSO were incubated in a 1536-well plate (782096, Greiner, Kremsmünster, Austria) at 37 °C for 24 h. The QDAβ-Aβ_42_ aggregates that formed in each well were observed by an inverted fluorescence microscope (TE2000, Nikon, Tokyo, Japan). Standard deviation (SD) values of fluorescence intensities of 40,000 pixels (200 × 200 pixels) around the central region of each well were measured by ImageJ software Ver 1.53b (NIH). The SD values, which were approximately proportional to the amount of aggregates [20,21,22], were plotted against the concentrations of added salad dressings to establish an inhibition curve.

### 2.4. Fluorescence Microscopy

Aggregates in the 1536-well plate were observed by an inverted fluorescence microscope (TE2000-S, Nikon) using a 4× objective lens (Plan Fluor 4×/0.13 PhL DL, Nikon) equipped with a color CCD camera (DP72, Olympus, Tokyo, Japan).

### 2.5. Transmission Electron Microscopy

Samples were deposited in 10 µL aliquots onto 200-mesh copper grids and negatively stained with 1% phosphotungstic acid at room temperature. Specimens were examined under an H-7600 TEM (Hitachi, Tokyo, Japan) at 60 kV.

### 2.6. ThT Assay

The ThT assay was conducted according to the method of Levine modified in our laboratory [18,21]. Statistical analyses between +ThT and −ThT samples were performed with EZR (Saitama Medical Center, Jichi Medical University, Saitama, Japan), which is a graphical user interface for R (The R Foundation for Statistical Computing) [23]. More precisely, it is a modified version of R commander designed to add statistical functions frequently used in biostatistics.

### 2.7. SDS-Polyacrylamide Gel Electrophoresis

SDS-polyacrylamide gel electrophoresis (SDS-PAGE) was performed using standard techniques. Soy sauce and raw soy sauce were heated by block incubator at 80 °C for 60 min before electrophorese. For dialysis protocol, soy sauce and raw soy sauce were dialyzed against distilled water. Distilled were changed three times for overnight. Then, the gel was silver-stained by staining kit (2D-SLVER STAIN II, COSMO BIO)

## 3. Results and Discussion

### 3.1. Evaluation of Aβ_42_ Aggregation Inhibitory Activity of Commercial Dressings by ThT Method

First, in order to assess whether 19 liquid salad dressings could be evaluated for activity by the ThT method, their absorbance spectra were measured using a Nanodrop 2000c (Thermo Fisher Scientific) (Figure 1). The excitation and emission wavelengths of ThT are 450 nm and 490 nm, respectively. As shown in Figure 1, only 3 of the 19 dressings (samples L, N, M) showed no absorbance of 1 or more at each wavelength. Most of the samples contained soy sauce as a raw material, and the color was black or brown depending on the content of soy sauce. In the 15 samples that showed absorption peaks at the ThT excitation and emission wavelengths, the absorption peak shifted to the right in proportion as the color became darker. This indicates that the evaluation of Aβ_42_ aggregation inhibitory activity of the liquid dressings using the ThT method was difficult because the absorption wavelength of almost samples overlapped with the excitation and emission wavelengths of ThT.

To confirm whether the evaluation using ThT method was performed correctly, the fluorescence intensity of five samples at a high concentration (40 *v*/*v*%) was measured in three conditions; +Aβ_42_ and +ThT, −Aβ_42_ and +ThT, −Aβ_42_ and −ThT (Figure 2). Soy sauce (sample A), soy sauce containing perilla (sample B), Japanese style dressing (sample C), oil dressing (sample D), and Chinese dressing (sample E) were selected and evaluated. Soy sauce (sample A) was used as the control for the other four samples. At 40 *v*/*v*% sample concentration, the fluorescence intensity of the −Aβ_42_ solution, −Aβ_42_ and −ThT solution were not significantly different from that of the solution containing Aβ_42_ and ThT. We confirmed that samples A and B showed higher fluorescence intensity than only Aβ_42_ and ThT sample in all conditions (negative control, black line). The fluorescence intensity of samples C, D, and E did not show a higher value than the negative control. The color and some components of the evaluated samples may affect the ThT method by absorbing the excitation or emission of ThT. If the sample solution without Aβ_42_ exhibits a higher fluorescence intensity than the negative control, it is difficult to determine whether the sample solution affects ThT. Therefore, the ThT method might not accurately evaluate the Aβ_42_ aggregation inhibitory activity of a commercial dressing.

### 3.2. Evaluation of Aβ_42_ Aggregation Inhibitory Activity of Commercial Dressings Using MSHTS Sysytem

Previously, we reported the real-time imaging of the Aβ_42_ aggregation process with a fluorescence microscope using QD nanoprobes (Figure 3A) and developed MSHTS system (Figure 3B) for Aβ_42_ aggregation inhibitors by applying this imaging method [19,20]. In the MSHTS system, QD-labeled Aβ co-aggregated with intact Aβ_42_, so that amyloid aggregates were observed by fluorescence microscopy. Since the emission wavelength of QD605 does not overlap with the absorption of almost dressings, it is less susceptible to substances that exhibit an inner filter effect. These aggregates (complex of Aβ_42_ and QD Aβ) caused an inhomogeneous distribution of fluorescence intensity in images, resulting in an increased standard deviation (SD) value estimated from the variation in fluorescence intensity of each pixel in the images. Therefore, we could estimate the effects of certain aggregation inhibitors by detecting changes in the SD value. Here, we evaluated the Aβ_42_ aggregation inhibitory activity of 19 commercial dressings using the MSHTS system (Figure 3C), and estimated the EC_50_ values from the SD value of each image (Table 1). In Table 1, EC_50_ values are sorted in ascending order of aggregation inhibitory activity, then each dressing was assigned a letter from A to S. As shown in Figure 3C, all commercial dressings almost completely inhibited Aβ_42_ aggregation at a concentration of 40 *v*/*v*%. Samples A to J completely inhibited aggregation even at 4 *v*/*v*%, whereas samples K to S formed small aggregates. At a concentration of 0.4 *v*/*v*%, a slight change in the shape of aggregates was observed in samples A to E. The EC_50_ value of the sample with the highest activity was 0.065 *v*/*v*%, and that with the lowest activity was 6.737 *v*/*v*%. There was an about 100-fold difference in activity between these 2 samples. Among the 19 dressings, only three samples (D, K, and N) contained plant oil while the other 16 samples were non-oil type dressings. We compared the mean value of EC_50_ value of non-oil and oil type (non-oil type: 1.330 ± 1.692 *v*/*v*%, oil type: 1.019 ± 0.946 *v*/*v*%) and performed a statistical analysis. There was no significant difference between non-oil type and oil type (Student’s *t*-test, *p* > 0.05). The activity did not depend on the presence of oil, suggesting that oil in the dressing did not affect the Aβ_42_ aggregation inhibitory activity. Next, to confirm whether Aβ_42_ aggregation was inhibited by the effect of the dressing, we observed the Aβ_42_ aggregates at a sample concentration of 0.04 *v*/*v*% using TEM (Figure 4). Sample A, which had high Aβ_42_ aggregation inhibitory activity, showed a significant decrease in Aβ_42_ aggregates compared to samples J and S. Sample J also significantly decreased aggregates than sample S. These results were consistent with the Aβ_42_ aggregation inhibitory activity calculated by the MSHTS system. Sample A, which showed the highest Aβ_42_ aggregation inhibitory activity, was soy sauce, a traditional Japanese liquid dressing. The remaining 17 samples, except for sample D, contained soy sauce. In other words, soy sauce evidently exhibited high Aβ_42_ aggregation inhibitory activity. Natto, a traditional Japanese food made of fermented soybeans, has antibacterial, as well as a soybean peptide with a neuroprotective effect [24]. It is possible that soybean-derived proteins found in soy sauce, and/or various metabolic products caused by soybean fermentation, may have a positive effect on AD.

### 3.3. Effect of Salt Concentration on Aβ_42_ Aggregation

In general, it is well known that the soy sauce contains a large amount of NaCl and that high salt concentration affects protein aggregation. In order to determine whether the Aβ_42_ aggregation inhibitory activity of soy sauce was due to NaCl, we examined the effect of NaCl concentration on the shape of Aβ_42_ aggregates (Figure 5A). We prepared a 4000 mM NaCl solution. Then, the solution was gradually diluted to six concentrations with five-fold dilutions. These 7 concentrations of NaCl solution was mixed with the 50 μM Aβ_42_ solution and observed using the MSHTS system. The formation of Aβ_42_ aggregates was slightly affected by 200 mM NaCl solution. At 1000 mM of NaCl, Aβ_42_ aggregates were fragmented, and at 2000 mM, abnormal aggregates such as large clumps were observed. The SD values obtained from images were gradually decreased from 2000 mM to 200 mM (Figure 5B). However, 0.32–40 mM NaCl did not affect the SD value, suggesting that Aβ_42_ aggregates were formed. The NaCl concentration in sample A soy sauce is 16.2% (2.77 M). Because the EC_50_ of sample A is 0.065 ± 0.020 *v*/*v*%, it was indicated that 1.8 mM NaCl is included. The NaCl concentration of the 0.4 *v*/*v*% and 0.04 *v*/*v*% solution samples is 11 mM and 1.1 mM, respectively. Therefore, the NaCl in 0.065 *v*/*v*% (EC_50_ value) soy sauce solution might not affect the formation of Aβ_42_ aggregates. In fact, as shown in Figure 3C, aggregation was inhibited in samples A to J when NaCl was included at 0.4 *v*/*v*%. Especially, 0.04 *v*/*v*% of sample A inhibited the aggregation formation. These results suggest that the Aβ_42_ aggregation inhibitory activity by soy sauce was due to not NaCl but other components.

### 3.4. Influence of Heating and Dialysis Treatment on Aβ_42_ Aggregation Inhibitory Activity

As shown in Figure 3, among the 19 dressings, soy sauce showed the highest activity when the MSHTS system was used. In fact, there are several types of soy sauce, which can be classified according to the sterilization method, the composition of the raw materials, the color, and salt concentration of the product. Among them, we focused on raw soy sauce that was not heat-sterilized. Since raw soy sauce is sterilized by filtration, it has a feature that compounds produced during the fermentation process and enzymes derived from microorganisms are not inactivated. Here, we evaluated Aβ_42_ aggregation inhibitory activity of raw soy sauce purchased from NIHON SYOYU KOGYO using the MSHTS system. The Aβ_42_ aggregation inhibitory activity of raw soy sauce was 0.0045 ± 0.0015 *v*/*v*% (Data not shown). This activity is about 15 times higher than that of soy sauce which showed the highest activity among the 19 dressings. Therefore, it is likely that the difference in the aggregation inhibitory activity of Aβ_42_ between soy sauce and raw soy sauce is caused by the raw material-derived protein and the microorganism-derived enzyme, which are lost by heating.

To examine whether proteins and/or low molecular weight compounds in soy sauce and raw soy sauce affect Aβ_42_ aggregation inhibitory activity, each sample was subjected to SDS-PAGE after heat treatment (80 °C, 60 min) and/or dialysis treatment and their band patterns were compared using silver staining (Figure 6A). As revealed by an SDS-PAGE gel (lanes 1 and 5), the property and amount of proteins in soy sauce and raw soy sauce differed. Bands were detected at 15 and 25 kDa in both samples while 30 and 100–200 kDa bands were detected only in raw soy sauce. Heat treatment did not change the banding pattern of soy sauce, but reduced the intensity of the100–200 kDa band of raw soy sauce (Figure 6A, from lanes 2 and 6). Dialysis treatment reduced the 10–15 and 25 kDa bands in both samples (Figure 6A, lanes 3 and 7). Heat treatment after dialysis reduced the band at about 25 kDa in soy sauce less than the dialyzed sample, and reduced the bands at about 15, 25 and 100–200 kDa in raw soy sauce (Figure 6A, lanes 4 and 8).

Next, using the MSHTS system, we assessed whether heating and dialysis treatment would affect the Aβ_42_ aggregation inhibitory activity of soy sauce and raw soy sauce (Figure 6B). Furthermore, the dialyzed soy sauce and raw soy sauce were also heated and analyzed. As shown in Figure 6C, the Aβ_42_ aggregation inhibitory activity of soy sauce was not changed by heating. This result was consistent with the SDS-PAGE banding pattern (Figure 6A, lanes 1 and 2). In addition, the activity of soy sauce decreased even after dialysis treatment, even more so when heat treatment followed dialysis. The Aβ_42_ aggregation inhibitory activity of raw soy sauce was reduced by about 30 times or by about half after heat treatment and dialysis, respectively. Dialysis treatment did not affect the 100–200 kDa band that was reduced by heat treatment, i.e., the 100–200 kDa protein did not contribute to the Aβ_42_ aggregation inhibitory activity of raw soy sauce. The heat treatment after dialysis decreased Aβ_42_ aggregation inhibitory activity by about 440 times, which was consistent with the SDS-PAGE result in which many bands were reduced (Figure 6A, lane 8). These results suggest that the Aβ_42_ aggregation inhibitor found in soy sauce and raw soy sauce is a low molecular weight compound that is removed by dialysis and a protein that is thermally denatured by heat treatment at 80 °C for 60 min. Since the Aβ_42_ aggregation inhibitory activity of raw soy sauce was greatly reduced after heat treatment, the proteins found in raw soy sauce are considered to be particularly important for the inhibition of Aβ_42_ aggregation.

In general, proteins and peptides are denatured by heating, then lose their physiological activity. It is possible that the presence or absence of heat treatment in the fermentation process may be involved in the physiological activity of dressings such as soy sauce, which contains Aβ_42_ aggregation inhibitory activity. The main raw materials of soy sauce, soy, and wheat, are decomposed into amino acids, peptides and saccharides in the manufacturing process. It is known that soybeans have many physiologically active ingredients such as soy protein and isoflavone [25,26,27], so these physiologically active ingredients are also present in soy sauce. Soy is widely applied to fermented foods such as miso and natto, Japanese traditional foods. Actually, physiological activity has also been reported for these foods. Miso extract suppresses Aβ-induced neuronal damage [16]. Genistein, one of isoflavone, mitigated Aβ deposition and neuroinflammation in mice [28]. It was reported that natto peptide exhibited antimicrobials effects and that nattokinase has amino residues playing a intramolecular chaperone [24,29] Further, vitamin K2 (menaquinone-7), which is abundant in natto, is an important factor in the synthesis of sphingolipids present in brain cell membranes that support cell signaling function and structure formation [30,31,32]. We are currently investigating the active compound in raw soy sauce using this analytical method. Soy sauce is used in many traditional dishes in Japan, and its effectiveness against AD would be significant for the prevention of this disease.

## 4. Conclusions

In this work, we evaluated Aβ_42_ aggregation inhibitory activity of 19 commercial liquid dressings using the MSHTS system. All tested dressings exhibited Aβ_42_ aggregation inhibitory activity, suggesting that the MSHTS system can be applied to processed food containing various impurities. Japanese traditional liquid dressings, soy sauce, exhibited the highest inhibitory activity. However, these findings are limited to in vitro conditions. The physiological activity of the dressings should be clarified through animal experiments, taking into account dynamics such as intestinal absorption and metabolism, particularly the permeability of the blood-brain barrier. Although there is a strong demand for functional food products that help maintain and improve brain function, it is not realistic to subject all food products to animal testing. We are confident that we can progress quickly to a second screening stage such as animal testing by using the MSHTS system as a first screening tool to discover food materials with high Aβ_42_ aggregation inhibitory activity. Recently, we confirmed that aggregation of various amyloid proteins Aβ_42_, tau, and α-synuclein could be visualized using nonlabelled QD and succeeded in the evaluation of aggregation inhibitory activity of RA [33]. MSHTS system using nonspecific binding of QD to amyloid proteins might bring speed and simplification of the screening of various foods using various amyloids. Furthermore, we expected that the combination of our previously developed automated-MSHTS system [21] and non-specific MSHTS system allows enormous, comprehensive screening of foods, thereby creating the potential for new approaches to overcome AD.

## Figures and Tables

**Figure 1 foods-09-00825-f001:**
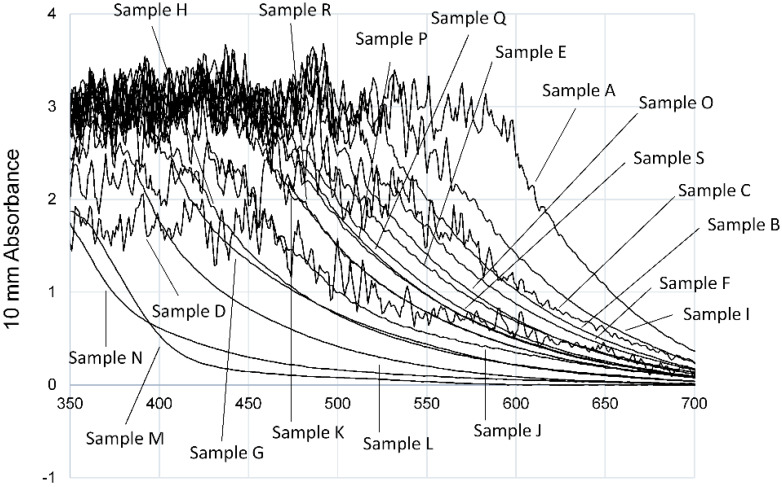
Absorbance of 19 commercial liquid dressings. Absorbance of the 19 commercial liquid dressings shown in Table 1 was measured. As for the absorption wavelength of dressings, three samples (L, M, N) do not show an overlap with the excitation (450 nm) and emission (490 nm) wavelengths of ThT. Two samples (H, G) overlap with the ThT excitation wavelength. The remaining 14 samples had an overlap with both excitation and emission wavelengths.

**Figure 2 foods-09-00825-f002:**
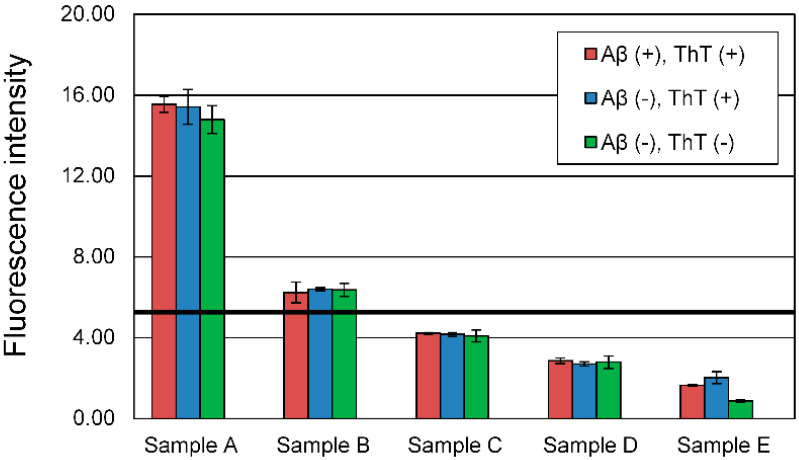
Effect of sample solution on ThT fluorescence intensity. Aβ_42_ (+), ThT (+): Aβ_42_ solution and ThT solution are mixed with sample; Aβ_42_ (−), ThT (+): only ThT solution is mixed with sample solution; Aβ_42_ (−), ThT (−): neither Aβ_42_ solution nor ThT solution are mixed and only the sample solution is used. As a negative control, the sample used was an assay buffer (10% EtOH, 1 × PBS) under the conditions of Aβ_42_ (+) and ThT (+) (black line), and its average value of absorbance was 5.28 (A.U.). There is no significant difference between +ThT (+/− Aβ_42_) and ThT (−) condition in all samples (One way ANOVA, *p* > 0.05).

**Figure 3 foods-09-00825-f003:**
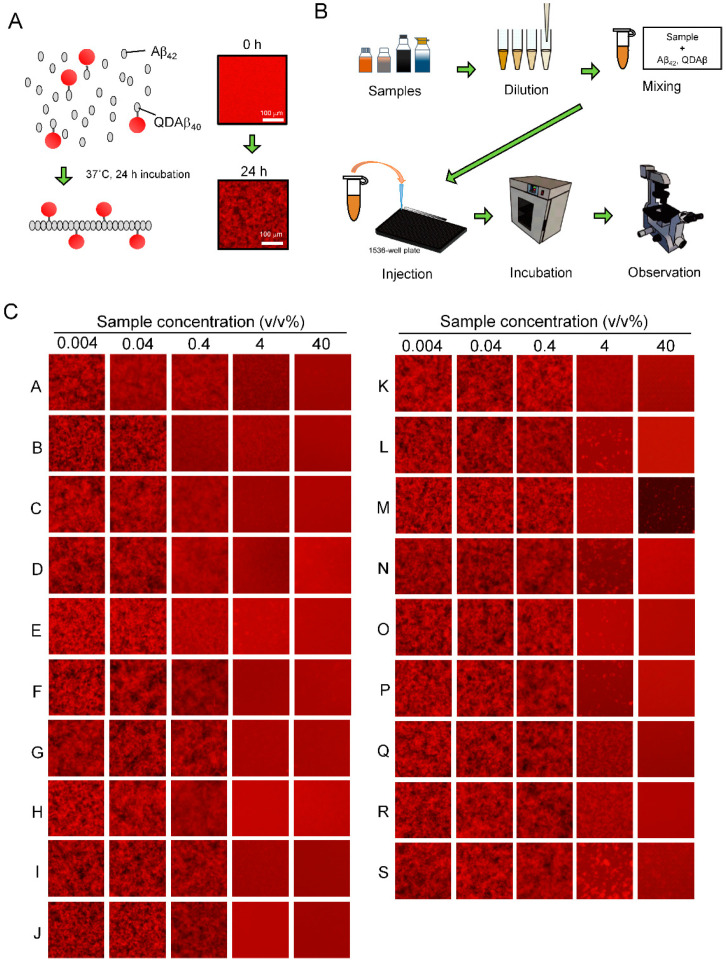
Evaluation of Aβ_42_ aggregation inhibitory activity using the MSHTS system. (**A**) Real-time imaging of Aβ_42_ aggregation using a quantum-dot nanoprobe using fluorescence microscopy. Aβ_42_ and QDAβ were mixed and incubated for 24 h at 37 °C. Co-aggregates of Aβ_42_ and QDAβ formed. (**B**) A scheme of the MSHTS system of Aβ_42_ aggregation inhibitors. (**C**) Fluorescence microscope image of concentration-dependent inhibition of Aβ_42_ aggregation of dressings observed by the MSHTS system. At a sample concentration of 4 *v*/*v*% or more, the brightness of the image was uniform, indicating that no Aβ_42_ aggregates formed. At a sample concentration of 0.04 *v*/*v*% or less, Aβ_42_ aggregates were observed in all samples. All images were captured using a conventional fluorescence microscope.

**Figure 4 foods-09-00825-f004:**
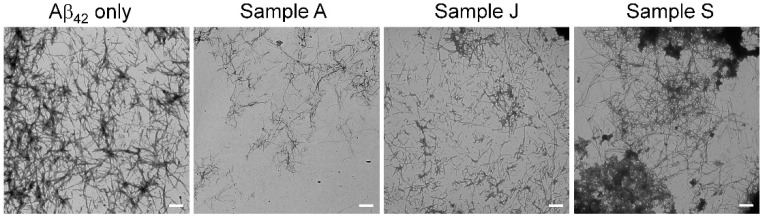
Electron microscopy images of Aβ_42_ aggregates. The Aβ_42_ solution was mixed with each sample (0.04 *v*/*v*%) and was incubated for 24 h at 37 °C. The images of Aβ_42_ aggregates were captured by TEM at 3000× magnification. Bars: 500 nm.

**Figure 5 foods-09-00825-f005:**
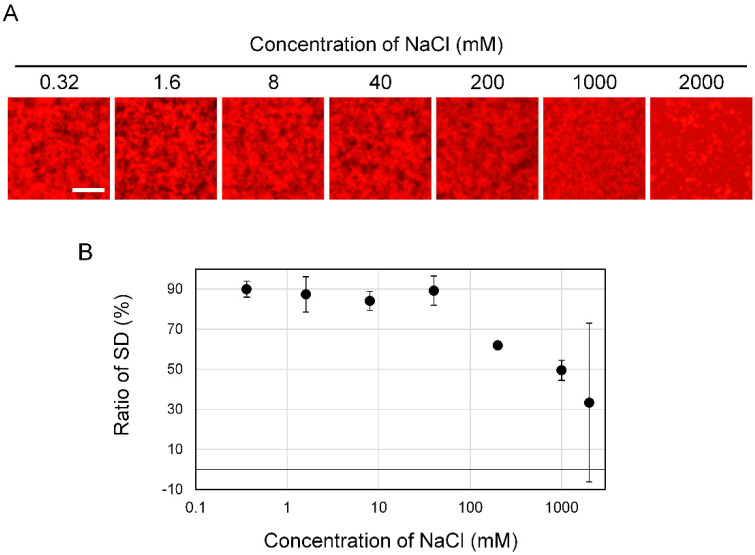
Influence of salt concentration on Aβ_42_ aggregation. (**A**) NaCl solution adjusted to each concentration and the Aβ_42_ solution were mixed then incubated for 24 h at 37 °C. Using the MSHTS system, the influence of NaCl concentration on the formation of Aβ_42_ aggregates was examined. At 2000 mM NaCl, Aβ_42_ and QDAβ were salted out. From their morphology, it is believed that these solids were not Aβ_42_ aggregates. At 1000 mM, aggregates started to form. At 40 mM or less, no significant effect was observed on the aggregates. From 40 to 0.32 mM, normal aggregates were formed. Bars: 100 μm. (**B**) Ratio of SD value at each NaCl concentration. At 40 mM or less, the SD value was not affected by NaCl concentration.

**Figure 6 foods-09-00825-f006:**
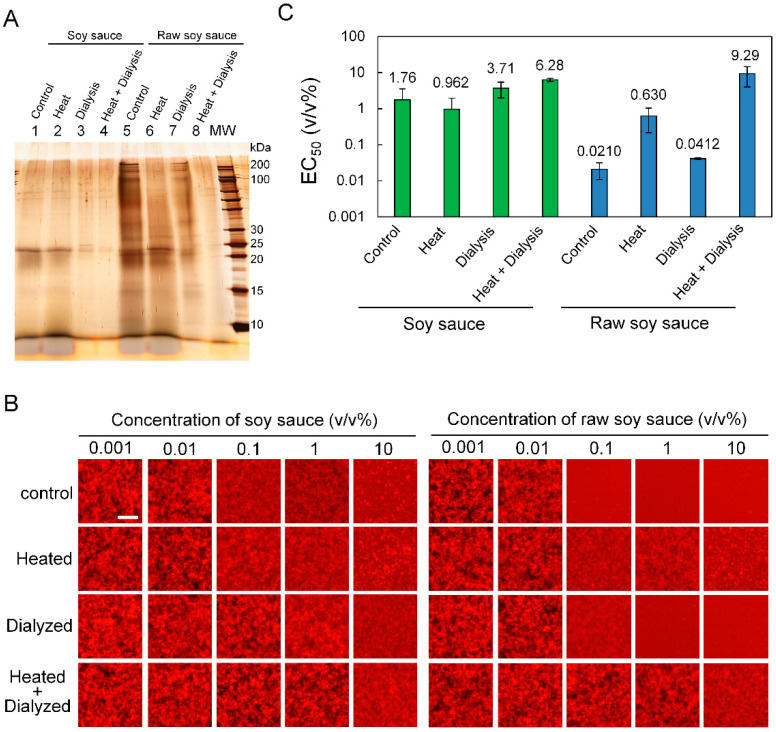
Influence of heat and dialysis treatment on Aβ_42_ aggregation inhibitory activity of soy sauce and raw soy sauce. (**A**) SDS-PAGE analysis demonstrated that heating and dialysis treatments changed the band pattern of soy sauce and raw soy sauce. Lanes 1–4: soy sauce; lanes 5–8: raw soy sauce. Lanes 1 and 5: control; lanes 2 and 6: heat treatment; lanes 3 and 7: dialysis treatment; lanes 4 and 8: dialysis and heat treatment. MW: molecular weight marker. (**B**) Fluorescence images of Aβ_42_ aggregates in each condition. Effect of heating and dialysis treatment on Aβ_42_ aggregation inhibition of soy sauce and raw soy sauce. All images were captured using a conventional fluorescence microscope. Bars: 100 μm. (**C**) Aβ_42_ aggregation inhibitory activity (EC_50_) of soy sauce and raw soy sauce after heating and dialysis calculated by the MSHTS system. Whereas the activity of soy sauce was not changed by heating, the EC_50_ of heated raw soy sauce was about 30 times higher than that of the control sample, suggesting that the activity was greatly reduced. In both soy sauce and raw soy sauce, EC_50_ was approximately double after dialysis.

**Table 1 foods-09-00825-t001:** EC_50_ values of 19 commercial dressing samples using MSHTS system.

Sample	EC_50_ (*v*/*v*%)	Oil Type
RA (positive control)	0.122 ± 0.034 (*w*/*v*%)	-
A	0.065 ± 0.020	Non-oil
B	0.094 ± 0.017	Non-oil
C	0.133 ± 0.021	Non-oil
D	0.227 ± 0.026	Oil
E	0.230 ± 0.026	Non-oil
F	0.334 ± 0.075	Non-oil
G	0.395 ± 0.130	Non-oil
H	0.413 ± 0.084	Non-oil
I	0.480 ± 0.101	Non-oil
J	0.508 ± 0.025	Non-oil
K	0.763 ± 0.607	Oil
L	1.350 ± 0.247	Non-oil
M	1.360 ± 0.590	Non-oil
N	2.067 ± 0.728	Oil
O	2.132 ± 1.473	Non-oil
P	2.150 ± 0.887	Non-oil
Q	2.313 ± 0.490	Non-oil
R	2.580 ± 0.173	Non-oil
S	6.737 ± 5.054	Non-oil

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
