# Peer review of "Evaluation of Amyloid β42 Aggregation Inhibitory Activity of Commercial Dressings by A Microliter-Scale High-Throughput Screening System Using Quantum-Dot Nanoprobes"

_foods, 2020, doi:10.3390/foods9060825_

Round 1

Reviewer 1 Report

The study is interesting. My suggestions are following:

  1. The aim is not defined in the abstract.
  2. The aim should be written at the end of the introduction.
  3. Lines 73-75 do not belong to that introduction part.
  4. Line 81. Where salad dressings and soy sauce brands were purchased, the country and the producer.
  5. Explain what is abbrevation PBS.
  6. Why certain statistical analysis was not applied on the study? Why not ANOVA in Figure 2?
  7. Would it be possible to apply PCA analysis for an example on the results from the Table 1. Graphical presentation with PCA would give more clear picture about the gained results.

Author Response

Reviewer #1

We thank the reviewer very much for attentive assessing our manuscript. 

1: The aim is not defined in the abstract.

2: The aim should be written at the end of the introduction.

As the reviewer pointed out in comment 1 and 2, we agree that it is not enough the explanation about aim of our study. In this study, we aim to elucidate whether the MSHTS system could be applied to processed foods such as salad dressings including various natural substances. To clarify our purpose, we added the relevant description in our abstract (Revised manuscript, Page 1, Line 15 to 17. Then, we also revised Introduction section to emphasize our purpose (Revised manuscript, Page 2, line 88 to 90).

3: Lines 73-75 do not belong to that introduction part.

As reviewer indicated, we agree that detailed results of study should not be included in the Introduction section. We have briefly revised the relevant parts (Revised manuscript, Page 2 to 3, line 91 to 93).

4: Line 81. Where salad dressings and soy sauce brands were purchased, the country and the producer.

According to the reviewer's comment, we described information of the country and the producer of used salad dressings and soy sauce in Material and Methods section (Revised manuscript, Page 3, line 99 to 102).

5: Explain what is abbreviation PBS.

As the reviewer indicated, we should explain about abbreviation of Phosphate-Buffered Saline (PBS). We described abbreviation of PBS (Revised manuscript, Page 3, Line 108). Further, we newly created abbreviation list of this manuscript (Revised manuscript, Page 12, Line 359).

6: Why certain statistical analysis was not applied on the study? Why not ANOVA in Figure 2?

As the reviewer indicated in comment 6, we agree that we should perform statistical analysis in Fig. 2. Using one way ANOVA analysis, we confirmed that there is no significant differences between +ThT and -ThT. Then, we described explanation about statistical analysis in legend of Fig. 2 (Revised manuscript, Page 5, Line 177 to 178). And, we mentioned about statistical analysis software, EZR in Material and Methods section (Revised manuscript, Page 3, Line 130 to 133).    

7: Would it be possible to apply PCA analysis for an example on the results from the Table 1. Graphical presentation with PCA would give more clear picture about the gained results.

We agree that we perform PCA analysis for Table 1 to support the understanding of readers. We newly created Fig. 4 to show EC50 ranking of salad dressings clearly.

Reviewer 2 Report

This study seems to be carefully performed, well written and presented. I only have one comment - it would be great if authors can add the perspective of the study pertaining to the applicability and broad impact of the study.  

Author Response

Reviewer #2

We would like to thank the reviewer for encouraging comment on our study.

This study seems to be carefully performed, well written and presented. I only have one comment - it would be great if authors can add the perspective of the study pertaining to the applicability and broad impact of the study. 

As a reviewer suggested, we should add the perspective of our study to emphasize the impact. In Conclusion section of revised manuscript, we mentioned about the perspective of our study in the future (Revised manuscript, Page 13 to 14, Line 342 to 348) as follows. Most recently, we reported that we succeeded evaluation of aggregation inhibitory activity on Tau and α-synuclein addition to Aβ42 using unlabeled QD (Lin et al., 2020, IJMS). We expected that nonspecific MSHTS system using unlabeled QD could be applied to wider screening of aggregation inhibitory candidates due to its simplicity. Moreover, we believed that combination of unlabeled method and automated methods (Sasaki et al., 2019, Scientific Reports) on MSHTS could contribute to the search for new lead compounds for the development of preventive and therapeutic drugs for proteinopathy.

Reviewer 3 Report

In this paper, the Aβ42 aggregation evaluated with activity of 19 commercial liquid dressings 297 using the MSHTS method. The paper is well written and well presented. However, I expect the author to clarify the main contribution of the work. In addition, the author should explain the main issues and challenges for current approach to provide research directions for other researchers.

Author Response

Reviewer #3

We are grateful for the positive feedback from reviewer. 

In this paper, the Aβ42 aggregation evaluated with activity of 19 commercial liquid dressings 297 using the MSHTS method. The paper is well written and well presented. However, I expect the author to clarify the main contribution of the work. In addition, the author should explain the main issues and challenges for current approach to provide research directions for other researchers.

As a reviewer indicated, we agree that description about perspective of our study is not enough. We added the explanation of next issue and approach in Conclusion section (Revised manuscript, Page 13 to 14, Line 342 to 348). In added region, we cited and mentioned our current work. Most recently, we reported that MSHTS using unlabeled QD could evaluate that aggregation inhibitory activity using not only Aβ42 but Tau and α-synuclein (Lin et al., 2020, IJMS). We expected that nonspecific MSHTS system could apply to screening of various food on other amyloid protein.

Reviewer 4 Report

Dr. Kuragano et al conducted a high-throughput screen of commercial dressings for amyloid aggregation using their MSHTS system. Traditional Japanese soy sauce is found to have a strong inhibitory effect. They ruled out the effect of sodium chloride and found other ingredients exhibit the inhibitory effects.

It is a nicely conducted study. I have a few minor comments:

1) in abstract, make it clear the EC50s are dilutions to avoid confusion. 

2) The sodium chloride experiment (Figure 5). I think it is important to provide the 100mM point. Because 4% soy sauce contains ~110mM NaCl, the inhibitory effects at ~100mM is critical to interpret the results. 

3) Figure 6A, add treatments for each lane above the gel image. Figure 6B, Y-axis title is confusing. It is noted as logEC50 but the ticks are not log values. 

4) Line 179-180, double check the sentence. Other minor language issues are also present.

Author Response

Reviewer #4

We appreciate the reviewer’s constructive criticism of our manuscript.

1: in abstract, make it clear the EC50s are dilutions to avoid confusion.

As a reviewer pointed out, we agree that it is not enough the explanation about EC50 value. In revised abstract, we explained the dressings were serially diluted in this study (Revised manuscript, Page 1, Line 17 to 18).

2: The sodium chloride experiment (Figure 5). I think it is important to provide the 100mM point. Because 4% soy sauce contains ~110mM NaCl, the inhibitory effects at ~100mM is critical to interpret the results.

Thank you for pointing out. We think that the explanation about the sodium chloride experiment (previous manuscript, Fig. 5) is not appropriate. In previous manuscript, Fig. 5 (Revised manuscript, Fig. 6), we examined the effect of NaCl concentration on MSHTS system. As shown in Fig. 3, the sample A, soy sauce exhibited aggregation inhibitory activity even at 0.04 v/v %. Actually, we estimated the EC50 value of sample A is 0.065 ± 0.020 v/v %. Because the NaCl concentration of used soy sauce is 16.2% (2.77 M), NaCl concentration of the 0.4% and 0.04% soy sauce is 11 mM and 1.1 mM, respectively. Further, NaCl concentration at EC50 of soy sauce is 1.8 mM. The SD values at these concentrations were not affected, indicating that aggregates formation was not inhibited. Therefore, we thought that important concentration range of NaCl is 0.32-40 mM and concluded that we can ignore the effect of NaCl in MSHTS assay. To sure the reader’s understand, we revised the explanation about the sodium chloride experiment (Revised manuscript, Page 9, Line 242 to 246).

3: Figure 6A, add treatments for each lane above the gel image. Figure 6B, Y-axis title is confusing. It is noted as logEC50 but the ticks are not log values.

Thank you for important advice to sure reader’s appropriate recognition. We added the treatment name for each lane above gel image (Revised manuscript, Fig. 7A) to support the understanding of reader. Then, we had mistaken in label title (previous manuscript, Fig. 5B; X axis and Fig. 6C; Y axis). We corrected each label title (Revised manuscript, Fig. 6B; X axis and Fig. 7C; Y axis).

4: Line 179-180, double check the sentence. Other minor language issues are also present.

We indeed had a mistake in the sentence reviewer indicated. Here, we suggested that amount of aggregates in sample J is less than sample S. We apologize and correct the relevant region (Revised manuscript, Page 6, Line 205 to 206).

Round 2

Reviewer 1 Report

I would suggest following corrections:

  1. no significant differences should be marked in the text in brackets (p>0.05).
  2. The figure of principal component analysis I do not see in the manuscript. Please provide the figure of principal component analysis.

Author Response

Reviewer #1

We thank the reviewer for critically and constructive suggestions.

1: No significant differences should be marked in the text in brackets (p>0.05).

As a reviewer indicated, we agree that we should mark no significant differences in the text in brackets. We described that P value was greater than 0.05 in the legend of Fig. 2 (Revised manuscript, Page 5, Line 181).

2: The figure of principal component analysis I do not see in the manuscript. Please provide the figure of principal component analysis

Thank you for important indication. However, we thought that at least two variable values are needed for the principal component analysis and creation of plotting graph. In table 1, we showed just EC50 value of each dressing with sample name (A to S), indicating that variable value is just one. We are afraid that we can hardly fathom reviewer's true meaning. If you have other idea for creation of graph, could you teach us specifically what kind of plotting graph should be created?